# Diallyl Disulfide Induces Chemosensitization to Sorafenib, Autophagy, and Cell Cycle Arrest and Inhibits Invasion in Hepatocellular Carcinoma

**DOI:** 10.3390/pharmaceutics14122582

**Published:** 2022-11-24

**Authors:** Ana Rita Thomazela Machado, Katiuska Tuttis, Patrick Wellington da Silva Santos, Alexandre Ferro Aissa, Lusânia Maria Greggi Antunes

**Affiliations:** 1Department of Clinical Analyses, Toxicology and Food Science, School of Pharmaceutical Sciences of Ribeirão Preto, University of São Paulo, Ribeirão Preto 14040-904, SP, Brazil; 2Department of Genetics, Ribeirão Preto Medical School, University of São Paulo, Ribeirão Preto 14040-904, SP, Brazil; 3Institute of Biomedical Sciences, Federal University of Alfenas, Alfenas 37130-001, MG, Brazil

**Keywords:** liver cancer, nutraceutical, *Allium sativum* L., DNA damage, nutrigenomics

## Abstract

Hepatocellular carcinoma is the seventh most common type of cancer in the world, with limited treatment options. A promising strategy to treat cancer is to associate chemotherapeutics and plant bioactive compounds. Here, we examined whether diallyl disulfide (DADS; 50–200 μM) and sorafenib (SORA; 8 μM), either alone or in combination, were toxic to hepatocellular carcinoma cells (HepG2) in vitro. We assessed whether DADS and/or SORA induced cell death (LIVE/DEAD assay and autophagy) and cell cycle changes (flow cytometry), altered expression of key genes and proteins (RT-qPCR and Western blot), and modulated tumorigenesis signatures, such as proliferation (clonogenic assay), migration (wound healing), and invasion (inserts). The DADS + SORA combination elicited autophagic cell death by upregulating LC3 and NRF2 expression and downregulating *FOS* and *TNF* expression; induced the accumulation of cells in the G1 phase which thereby upregulated the *CHEK2* expression; and inhibited invasion by downregulating the *MMP2* expression. Predictive analysis indicated the participation of the MAPK pathway in the reported results. The DADS + SORA combination suppressed both cell invasion and clonogenic survival, which indicated that it dampened tumor growth, proliferation, invasion, and metastatic potential. Therefore, the DADS + SORA combination is a promising therapy to develop new clinical protocols.

## 1. Introduction

Primary liver cancer is the seventh most diagnosed cancer type and the fifth leading cause of cancer mortality worldwide. The treatment of hepatocellular carcinoma involves a complex decision-making process because patients are often diagnosed at advanced stages, when potentially curative therapies such as resection, ablation, and transplantation have limited effectiveness [1,2,3]. Sorafenib (SORA), an oral multikinase inhibitor, is used to treat cancer at advanced stages and acts by inducing cancer cell death and altering the tumor microenvironment [4]. However, this drug can be administered only to patients with preserved liver function, does not completely stop disease progression in most patients, and extends their survival by two to three months [3]. Due to the limited options to treat liver cancer, associated with poor prognosis of the disease, there is a critical need for new therapies to improve patients’ survival and quality of life [5].

Natural products are excellent sources in the search for novel chemotherapeutic and chemopreventive agents and have already provided several currently used chemotherapy drugs [6,7,8]. Biologically active natural products tend to be well tolerated, readily available, and inexpensive [9] and can act synergistically with the chemotherapeutic drugs used in the clinic [10,11]. Diallyl disulfide (DADS), a major oil-soluble compound derived from garlic, is a multi-target antitumor agent that induces cellular processes, including cell cycle arrest, growth inhibition, differentiation, and apoptosis in many cancer types [12,13]. The antitumor mechanisms of garlic extracts include inhibition of cell growth and proliferation, regulation of carcinogenic metabolism, stimulation of apoptosis, prevention of angiogenesis, invasion, and migration, and the reduction in side effects [14]. DADS is an effective modulator of protein kinases/phosphatases that suppresses the tumor initiation and promotion phases [12], activates metabolizing enzymes that detoxify carcinogens, suppresses DNA adduct formation, induces apoptosis, and inhibits angiogenesis, histone modification, and tumor invasion [15,16].

DADS exerts antitumor activity in vitro in breast [17], colon [18], prostate [19], and liver [20] cancer cells. The promising results in different tumor cell lines have stimulated further research on new approaches for applying this bioactive compound to treat patients with hepatocellular carcinoma. In this sense, the present study examined whether DADS, either alone or in combination with the chemotherapeutic agent SORA, induced the cell death of HepG2 cells via autophagy, suppressing cell migration and invasion, and modulated genes and protein expression. To analyze the selectivity of action, some of their effects were also assessed in human umbilical vein endothelial cells (HUVECs). HUVEC cells are a non-tumor cell model often used to compare drug action in HepG2 cells [21,22,23,24,25,26].

## 2. Materials and Methods

### 2.1. Chemicals

Diallyl disulfide (DADS; CAS 2179-57-9), resazurin (CAS 62758-13-8), trypan blue (CAS 72-57-1), Tris (CAS 77-86-1), dimethyl sulfoxide (DMSO; CAS 67-68-5), ethylenediamine tetraacetic acid (EDTA, CAS 60-00-4), Triton X-100 (CAS 9002-93-1), and methyl methanesulfonate (MMS; CAS: 66-27-3) were obtained from Sigma-Aldrich (Saint Louis, MO, USA). Sorafenib (SORA; CAS 284461-73-0) was purchased from Santa Cruz Biotechnology (Dallas, TX, USA). RPMI 1640 culture medium, DMEM, antibiotic mix (penicillin/streptomycin/neomycin), fetal bovine serum, TrypLE™ Express, low melting point agarose (CAS: 39346-81-1), normal melting point agarose (CAS: 9012-36-6), and the Annexin V-FITC/PI assay kit (cat. #V13242) were acquired from Thermo Fisher Scientific (Waltham, MA, USA). GelRed™ dye (Biotium, Hayward, CA, USA) was obtained from Uniscience (São Paulo, SP. Brazil). Giemsa was purchased from Merck (Rockville, MD, USA). Doxorubicin hydrochloride (50 mg/10 mL) was acquired from Eurofarma (São Paulo, SP, Brazil).

### 2.2. Cell Lines and Culture Conditions

The human hepatocellular carcinoma cell line HepG2 was obtained from Cell Bank of Rio de Janeiro (BCRJ, Cat. No. 0103). The human umbilical vein endothelial cell line HUVEC was obtained from ATCC (American Type Culture Collection, Cat. No. CRL-4053). The cells were manipulated in a Class II type A1 laminar flow hood (Veco, Campinas, Brazil) and kept in a Forma Series II, Water Jacket CO_2_ Incubator (Thermo Fisher Scientific—Waltham, MA, USA) at 37 °C, under 5% CO_2_ and 95% humidity. The cells were cultured in DMEM (HepG2) and RPMI 1640 (HUVEC) culture medium, supplemented with 4 mM L-glutamine, 10% fetal bovine serum, and 1% antibiotics, according to the procedures reported by Bal-Price and Coecke [27]. 

### 2.3. Protocol Design

Confluent cells were treated with SORA and DADS, either alone or in combination, for 2 to 72 h, depending on the experiment. SORA and DADS were added simultaneously in the combined treatment. We tested SORA at 8 µM—which correlated with its plasma concentration [28,29,30]—and DADS at 50, 100, and 200 µM. Complete medium was used as a negative control, 0.25% DMSO as a vehicle control, and 300 µM MMS as a positive control. 

The concentrations used were selected using previous studies that identified significant biological effects in similar conditions. For DADS, the middle concentration tested of 100 µM was based on the studies that reported the induction of apoptosis [20,31] and the inhibition of proliferation in hepatocarcinoma cells [32]. The concentration of 8 µM of SORA was chosen according to the plasma concentration in humans after the administration of two 400 mg tablets per day [28,29].

### 2.4. LIVE/DEAD Assay

Cell viability was assessed using the LIVE/DEAD™ assay kit (Thermo Scientific; Canoga Park, CA, USA). HepG2 and HUVEC cells were seeded (5 × 10^3^ cells/well) in 96-well black plates, stabilized for 24 h, and treated for 72 h. After treatment, the cultures were washed with PBS and stained according to the manufacturer’s recommendations. Calcein AM retained in living cells produces an intense green fluorescence, while ethidium homodimer (EthD-1) enters cells with damaged membranes and produces a red fluorescence in dead cells. Fluorescence was recorded in a Synergy H1 plate reader (BioTek^®^; Winooski, VT, USA) with excitation/emission filters of 485/530 and 530/645 nm for calcein and EthD-1, respectively. The fluorescence values of the negative control were considered as 100% cell viability, and the results were expressed as percentage (%) of viable cells. Images were captured using the Leica DMI 6000B Fluorescence Microscope (Leica; Wetzlar, Hessen, Germany). 

### 2.5. Clonogenic Survival Assay

The assay was performed according to the protocol reported by Franken et al. [33]. HepG2 cells (5 × 10^4^ cells/well) were seeded in a 24-well cell culture plate, incubated for 24 h at 37 °C under 5% CO_2_, and treated for 72 h. Next, the cells were washed with PBS, trypsinized, and counted. One hundred fifty cells were seeded in 6-well plates in duplicate. After 12 days, the cells were washed with PBS, fixed in methanol:acetic acid:water (1:1:8 *v*/*v*) for 30 min, and stained with Giemsa:phosphate buffer (1:20 *v*/*v*) for 25 min. The cells were washed with distilled water and counted in a hemocytometer. Colonies containing 50 cells or more were considered viable. Plating efficiency (PE) and survival fraction (SF) were calculated as follows: PE = (counted cells/seeded cells) × 100; SF = (treated colonies/seeded colonies) × PE. 

### 2.6. Autophagy-Induced Cell Death

The assay was performed as specified by the Autophagy Assay Kit’s manufacturer (ab139484; Abcam, Cambridge, UK). The HepG2 cells (1 × 10^4^ cells/well) were seeded in a 96-well plate, stabilized for 24 h, and treated for further 24 h. Fluorescence was recorded in a Synergy 2 spectrophotometer (BioTek^®^; Winooski, VT, USA) with excitation/emission filters of 463/534 nm (green) and 350/461 nm (blue). The green fluorophore labels the autophagic vesicles that contain the LC3 protein while the blue fluorophore labels the cell nucleus. Fluorescence values are expressed as the autophagy ratio (ratio of green to blue fluorescence values) (Tusskorn et al., 2019). Images were acquired using the Leica DMI 6000B Fluorescence Microscope (Leica; Wetzlar, Hessen, Germany). 

### 2.7. Analysis of Cell Cycle Phase

The cell cycle was analyzed as described by Pozaroswki and Darzynkiewicz [34]. Briefly, the HepG2 cells were seeded in a 6-well plate (1 × 10^6^ cells/well), incubated for 24 h at 37 °C, under 5% CO_2_, and treated for 72 h. Next, the cells were trypsinized, washed, suspended in 0.5 mL of PBS, and gently aspirated several times to obtain a homogenous suspension. The cells were fixed in 4.5 mL of 70% ethanol solution for 2 h at 4 °C, washed, suspended in the staining solution—0.1% Triton X-100 (*v/v*), 5 μg/mL PI, and 100 μg/mL DNase-free RNase—and incubated for 30 min, at room temperature, in the dark. The cells were analyzed in a FACSCanto flow cytometer, and the resulting distribution of cell cycle phases in 10,000 events was analyzed using the ModFit LT software (Verity Software House—Topsham, ME, USA).

### 2.8. Cell Migration

The HepG2 cells (2 × 10^4^ cells/well) were grown to 90% confluence in a 24-well plate. Then, a scratch was made on the adherent cell monolayer using a sterile tip. Wound images were recorded with a camera coupled to an inverted microscope (Zeiss-Primovert, Zeiss, Oberkochen, Germany), at 0, 6, 12, 24, and 48 h of incubation. The method calculated the change in wound area over time as a percentage of wound closure, which was determined as the percentage of migration, using Image J software 1.53i version (National Institutes of Health, Bethesda, MD, USA) and the following equation: cell migration (%) = [(TA 0 h − TA 24 h)/(TA 0 h)] × 100, where TA is the total area.

### 2.9. Cell Invasion

Invasion assays were performed using Greiner Bio-One inserts (Kremsmünster, Austria). Matrigel^®^ was added to the top of the porous membranes (pore size, 8 μm). After overnight incubation at 37 °C, the coated inserts were then hydrated with culture medium and 5 × 10^5^ HepG2 cells. The lower compartments were filled with 10% fetal bovine serum in culture medium. The cells were treated for 24 h. The cells that invaded the bottom of the insert were fixed in 70% ethyl alcohol and stained with 0.1% crystal violet. Invasion was determined by counting an average number of cells from four different fields with the aid of an inverted microscope (Zeiss-Primovert), and the invasion rates were expressed as the ratios between the values of the treated group relative to the negative control.

### 2.10. Gene Expression

RNA was extracted using the PureLink RNA Mini Kit (Invitrogen; Carlsbad, CA, USA) following the manufacturer’s recommendations and was reverse transcribed using the High Capacity cDNA Reverse Transcription Kit. The samples were treated with DNase I (1 U/µL) for DNA removal. A quantitative PCR was performed with Power SYBR^®^ Green Master Mix in a Step One Plus system (Applied Biosystems, Carlsbad, CA, USA), using pre-designed KiCqStart^®^ SYBR^®^ Green Primers (Appendix A) for the gene expression analysis (Sigma-Aldrich): *ACTB*, *CHEK2*, *FOS*, *GAPDH*, *HPRT1*, *MMP2*, and *TNF*. The relative expression of each gene was calculated by the 2^−ΔΔCt^ method, using the reference genes *ACTB*, *HPRT1*, and *GAPDH* for normalization.

### 2.11. Protein Expression

For LC3 expression, the HepG2 cells (2 × 10^6^ cells) were seeded in 75 cm^2^ cell culture flasks and treated for 24 h. The protein was extracted with RIPA buffer supplemented with 1% protease and 0.1% phosphatase. The protein was quantified using the Pierce™ BCA Protein Assay kit from Thermo Fisher Scientific (Waltham, MA, USA). Then, 2× Laemmli sample buffer (Sigma-Aldrich, Saint Louis, MO, USA) was added to the samples (1:1) and boiled at 95 °C for 5 min. The proteins present in the samples were separated by SDS-polyacrylamide gel electrophoresis and transferred to a nitrocellulose membrane (Bio-Rad, Hercules, CA, USA). Non-specific binding was blocked by incubating the membrane with 5% skim milk in TBS-T (0.1% Tris Buffer Saline with Tween 20) for 1 h. After blocking, the membrane was incubated overnight at 4 °C with primary antibodies (Appendix A), washed with TBS-T, and subsequently incubated with HRP-conjugated secondary antibody for 1 h at room temperature. The membrane was washed again with TBS-T, and the secondary antibodies were detected using ECL (GE Healthcare, Chicago, IL, USA). The protein bands were visualized using the ChemiDoc™ system (Bio-Rad, Hercules, CA, USA).

### 2.12. Analysis of the Compound–Gene–Protein Interaction Network

The free online platform STITCH was used for the bioinformatics analysis of correlation among the compounds, genes, and proteins. Network analysis was performed using the “analysis” command at the end of the page by downloading the files in the Biological Process, Molecular Function, Cellular Component, KEGG pathways, PFAM protein domains, INTERPRO protein domains, and resources [35]. We analyzed the four genes (*CHEK2*, *FOS*, *MMP2*, and *TNF*) and two proteins (LC3 and NRF2) with altered expression in the RT-qPCR and Western blot.

### 2.13. Statistical Analysis

The results were expressed as the mean ± standard deviation of the triplicate of three independent biological experiments. Data from the isolated experiments were checked for normality using the Kolmogorov–Smirnov Test and compared using one-way analysis of variance (ANOVA) followed by the Tukey’s multiple comparison post-test. Data from the combined treatments were compared using two-way ANOVA followed by the Tukey’s multiple comparison test, where distinct letters indicated a significant difference within the same group. Statistical analysis of the data from the combined treatments relied on the comparison among the groups treated with DADS, SORA, and DADS + SORA, without intergroup comparisons. To validate the positive control, the data from the positive and negative controls were compared using the Student’s *t* test. For gene expression and Western blot assays, statistical analysis was performed using the Student’s *t* test (*p* < 0.05) associated with relative expression compared with the control group. All data were analyzed with the aid of the GraphPad Prism 8 software (La Jolla, CA, USA). *p* < 0.05 was considered statistically significant for the parameters analyzed. 

## 3. Results

### 3.1. DADS and SORA Combination Effectively Induces Cytotoxicity in HepG2 and HUVEC Cells

Figure 1A depicts the representative immunofluorescence microscopy images of the HepG2 cells treated with DADS and SORA, either alone or in combination. The merged images represent the superposition of the dead and live cells. All the DADS concentrations tested (50, 100, and 200 µM) were cytotoxic to HepG2 cells. SORA (8 µM) was as cytotoxic as 50 µM DADS (Figure 1B). The combination of DADS at all concentrations tested and 8 µM SORA enhanced SORA cytotoxicity. Figure 1C depicts representative immunofluorescence microscopy images of HepG2 cells treated with DADS and SORA in combination. The combination of 8 µM SORA and 200 µM DADS induced cytotoxicity the most effectively (Figure 1D). 

Figure 2A depicts representative images of the HUVEC cells treated with DADS and SORA, either alone or in combination. The three DADS concentrations tested (50, 100, and 200 µM) were cytotoxic to the HUVEC cells. DADS at 200 µM reduced cell viability the most effectively and was as cytotoxic as 8 µM SORA (Figure 2B). Figure 2C depicts representative images of HUVEC cells treated with DADS and SORA in combination. Compared with SORA alone, SORA cytotoxicity was enhanced when associated with the highest DADS concentration tested (200 µM) (Figure 2D).

### 3.2. DADS and SORA Combination Decreases Survival Fraction in HepG2 and HUVEC Cells

In addition to the acute cytotoxicity of the DADS and SORA combination on the HepG2 cells, we also wanted to check their effects on the proliferation of cells by a clonogenic survival assay (Figure 3). Compared with the negative control, DADS at 100 and 200 μM and SORA at 8 μM decreased proliferation of the HepG2 cells, with median survival fractions of 0.71, 0.57, and 0.21, respectively (Figure 3A). All the DADS concentrations tested decreased the proliferation of HUVEC cells in a concentration-independent manner and as effectively as 8 μM SORA. Treatment with DADS at 50, 100, and 200 μM and SORA resulted in survival fractions of 0.66, 0.64, 0.62, and 0.58, respectively (Figure 3C).

Compared with DADS alone, the combined treatment resulted in a lower survival fraction in the HepG2 cells, which was similar to the survival fraction of the cells treated with SORA alone. It means that SORA mediated the reduction in the survival fraction (Figure 3B). 

SORA alone (8 µM) reduced the survival fraction more strongly than in combination with 50 μM of DADS in the HUVEC cells. SORA associated with the other DADS concentrations reduced the survival fractions as effectively as the compounds alone (Figure 3D). The survival fraction of the HUVEC cells was higher than that of the HepG2 cells, at all the concentrations tested. As the data analysis did not detect cell selectivity, we did not use HUVEC cells in the subsequent experiments.

### 3.3. DADS and SORA Combination Modulates Autophagy-Mediated Cell Death

Cells under stress conditions such as the perturbation by pharmacological agents can trigger cellular mechanisms, such as autophagy, to maintain cell homeostasis. Conversely, autophagy has also been associated with autophagic (“type 2”) cell death in response to destructive conditions [36]. For this reason, we asked whether the mechanisms of cell death observed in the previous assays were related to autophagy by measuring the levels of LC3, a known marker of autophagy [37]. SORA and DADS, either alone or in combination, markedly augmented the LC3 levels in the HepG2 cells (Figure 4A). DADS at all the concentrations tested (50, 100, and 200 µM) increased the autophagy ratio in HepG2 cells. SORA (8 µM) and DADS (200 µM) alone similarly augmented the autophagy ratio (Figure 4B), but their combination increased the autophagy ratio the most effectively. The combination of 50 µM DADS and 8 µM SORA elicited an autophagy ratio similar to that induced by DADS alone. The combination of 100 µM DADS and 8 µM SORA induced an autophagy ratio that differed from that elicited by the other associated treatments, but it was lower than that induced by DADS alone (Figure 4C).

### 3.4. DADS and SORA Combination Modulates Cell Cycle

As the DADS and SORA combination decreased the survival fraction of the HepG2 cells, we asked how this inhibition of cell proliferation was associated with the cell cycle (Figure 5A,B). SORA alone decreased the percentage of cells in the G1 phase when compared to the negative control and the single treatments with DADS (Figure 5C). The combination of SORA and DADS at 50 and 100 µM showed similar percentages of cells in G1, both higher than SORA alone. The combination of SORA and DADS at 200 µM, however, showed a decreased percentage of cells in G1 compared with the concentrations of 50 and 100 µM, but it was also higher when compared with SORA alone (Figure 5C). Following the same trend, the combination of SORA and DADS at 50 and 100 µM showed similar percentages of cells in G2, both lower than SORA alone. The combination of SORA and DADS at 200 µM showed an increased percentage of cells in G2 compared with the concentrations of 50 and 100 µM, but it was also lower when compared with SORA alone (Figure 5D). Regarding the population of S-phase cells, all the combinations of SORA and DADS showed a similar percentage of cells in S phase, all of them higher than SORA alone (Figure 5E).

### 3.5. DADS and SORA Combination Effectively Suppresses Cell Migration and Invasion 

In addition to killing and inhibiting the proliferation of tumor cells, it is interesting that a drug also inactivates the potential of the tumor cell to migrate to other locations [38]. For this reason, we evaluated the capability of the combination of DADS and SORA to interfere with the migration and invasion of HepG2 cells (Figure 6 and Figure 7). Compared with the control, all the DADS and SORA concentrations tested significantly reduced the migration of HepG2 cells after 24 h, with no difference among the compounds (Figure 6A,B). The two compounds in combination inhibited the migration of HepG2 cells to the same extent as the compounds alone (Figure 6A,C).

There were representative fields of invasive HepG2 cells on the membrane surface after treatments. SORA and DADS at 100 µM and 200 µM alone decreased the cell invasion rate (Figure 7B). SORA alone induced a lower invasion rate than the 8 µM SORA combined with DADS at 50 or 100 µM, but it inhibited invasion at the same extent as 8 µM SORA combined with the highest DADS concentration tested (200 µM) (Figure 7C).

### 3.6. DADS and SORA Combination Induces Expression of Genes and Proteins Related to Cell Death Pathways

As the compounds inhibited cell proliferation and altered the cell cycle profile of the HepG2 cells, we asked whether these effects were influenced by the genes associated with cell proliferation and the cell cycle. CHK2 is a cell cycle checkpoint regulator and putative tumor suppressor [39]. *CHEK2* mRNA levels were increased in the combined treatment when compared with the cells treated with SORA alone (Figure 8A). The mRNA levels of *FOS*, a proto-oncogene associated with cell proliferation, differentiation, and transformation [40], were decreased in the combined treatments when compared with the control and DADS alone, but increased when compared with SORA alone. *TNF* stimulates the survival and proliferation of malignant cells [41] and has been associated with autophagy in cancer cells [42]. Compared with DADS alone, 50 µM DADS combined with 8 µM SORA downregulated the relative expression of *TNF* (Figure 8A). 

Tumor invasion and metastasis can be mediated by enzymes that degrade the components of the extracellular matrix [43]. As the combination of DADS and SORA inhibited the migration and invasion of HepG2 cells (Figure 6 and Figure 7), we measured the mRNA levels of Matrix metalloproteinase 2 (*MMP2*), which has been associated with cell migration in cancer [44]. Compared with SORA alone, 50 µM DADS alone or in combination with 8 µM SORA downregulated the relative expression of *MMP2*.

Autophagy was increased by the combined treatment (Figure 4). We explored this mechanism by measuring the protein levels of LC3, a marker protein for autophagic activity [37]. The LC3 protein levels were augmented in all groups when compared with the control; the treatment with DADS and SORA in combination increased the LC3 protein levels more strongly than the compounds alone (Figure 8B,C). NRF2 is a master regulator of the antioxidant response and is responsive to DADS [45]. Moreover, a loss of *Nrf2* is associated with a reduction in autophagy [46]. We observed an increase in NRF2 expression in the cells treated with DADS alone or in combination with SORA when compared to the control cells (Figure 8B,C).

In order to identify the molecular mechanisms underlying the effects of DADS and SORA on cell death, we used the STITCH database to further query the relevant connections between SORA, DADS, and the altered proteins (Figure 9). STITCH is a database of protein–chemical interactions that integrates many sources of experimental and manually curated evidence with text-mining information and interaction predictions. The platform used validated the experiments, the other databases, and the literature. We submitted the four genes and two proteins obtained from RT-qPCR and Western blot. The predicted interaction network demonstrated that the interactions among the compounds, genes (*CHEK2*, *FOS*, *MMP2*, and *TNF*), and proteins studied (LC3 and NRF2) formed a biologically connected group (Figure 9) with more interactions than expected (observed/expected 41/16) (*p* < 0.001), a clustering coefficient equal to 0.839, and signaling pathways in cancer and regulation of cell death by apoptosis mainly associated with MAPK cell signaling (http://stitch.embl.de/cgi/network.pl?taskId=nv9TPmaTyMjt (accessed on 30 August 2022)). 

## 4. Discussion

The liver tumor environment is complex and multivariate and limits the success of monotherapies [47]. Notably, drug resistance is frequent and poses the main barrier in the systemic treatment of hepatocellular carcinoma. In this sense, it is urgent to seek novel compounds or therapeutic approaches to threat this disease [48]. The combination of bioactive molecules and SORA may improve their sensitivity and cytotoxicity [10,11,48,49,50,51,52]. There are several bioactive molecules with known anti-cancer activity, such as DADS. DADS decreases the risk of carcinogen-induced cancer in experimental animals and suppresses the proliferation of various types of cancer cells [12].

In this work, DADS and SORA alone decreased the viability of HepG2 cells, at all concentrations tested, while their combination enhanced cytotoxicity. The literature reports that DADS at 100–400 μM decreased approximately 20–30% of cell viability after 24 h of treatment [31,53]. SORA at concentrations equivalent to its human plasma levels decreases the cell viability of HepG2 cells [28,29].

The compounds alone also decreased the cell viability of HUVEC cells; only 8 µM SORA combined with the highest DADS concentration tested (200 µM) decreased cell viability more strongly than the compounds alone. These findings agree with the literature reports that 50 and 100 µM DADS reduce the cell viability of human chondrocytes (C28/I2) by nearly 20% [54].

We used the clonogenic cell survival assay to examine whether DADS and SORA in combination suppressed the ability of cancer cells to proliferate endlessly, i.e., impaired their reproductive capability to form a large colony or clone [55]. DADS alone decreased cell proliferation in HepG2 to the same extent as SORA alone. SORA alone or in combination with DADS similarly affected cell proliferation, indicating the lack of additive effect. These treatments also affected the viability and proliferation of HUVEC cells but to a lesser extent. For this reason, we used only HepG2 cells in the next experiments to investigate the mechanism by which the compounds in combination induced tumor cell death.

The cell’s ability to migrate and invade tissues is essential for many physiological processes, including embryonic development, wound repair, tumor invasion, angiogenesis, and metastasis [38]. Several studies have indicated that actin cytoskeleton reorganization is the basis of the migration, adhesion, and invasion of tumor cells [56]. Treatment with SORA and DADS alone suppressed cell migration to the same extent as both compounds in combination at low concentrations. Only the highest DADS concentration tested in combination with SORA decreased cell invasion more effectively than the compounds alone. Magnolol, a bioactive compound extracted from *Magnolia officinalis* bark, acted synergistically with sorafenib to dampen tumor cell growth, the expression of anti-apoptotic proteins, and migration/invasion in HCC cells, when compared with sorafenib alone [57].

The mechanisms underlying the antimetastatic activity of DADS are not completely elucidated yet. Non-cytotoxic DADS concentrations (<68 μM) block the invasion and migration of esophageal carcinoma cells OE19 by downregulating the expression of the *MMP2* and *MMP9* metalloprotease genes [58]. The inhibition of *MMP2* decreases cellular migration in in vitro models of retinoblastoma [44]. This mechanism may be correlated with the decreased *MMP2* expression induced by DADS and SORA in combination, when compared with DADS alone. Matrix metalloproteinases can disrupt the balance of extracellular matrix degradation and thus promote metastasis [43]. Matrix metalloproteinase 2 (*MMP2*) can degrade most components of the extracellular matrix, and it is widely accepted that the effect of MMP2 on the extracellular matrix is closely associated with tumor invasion and metastasis [43]. We observed a decrease in the relative expression of the *MMP2* of the associated treatment, when compared to DADS and SORA alone, which was accompanied by an inhibition of the migration and invasion of HepG2 cells.

The combined treatment with DADS and SORA also downregulated the expression of *FOS*, which mediates the regulation of cell proliferation, differentiation, and transformation [59]. Cell cycle regulation is an important aspect of cell proliferation. In our study, DADS and SORA in combination induced the accumulation of cells in the G1 phase when compared with the compounds alone; this finding is in line with the increased expression of *CHEK2*, which has been correlated with the block of the cell cycle in the G1 phase [60,61]. We may suppose that the low *FOS* expression followed by the induction of the cell cycle arrest blocked the process and inhibited cell proliferation, as demonstrated in the cytotoxicity assays.

TNF-α is one of the most important inflammatory mediators of the cancer-associated inflammatory networks. Previous studies have reported that TNF-α is positively associated with high-grade tumors and predicts poor survival in patients with hepatocellular carcinoma [62,63]. TNF-α expression is related to SORA resistance in hepatocellular carcinoma cells [64] and the regulation of autophagy levels [42]. Autophagy is an autodigestive process that degrades cellular organelles and proteins and plays an important role in maintaining cell homeostasis against environmental stress [36]. Autophagy induction can suppress TNF-α release and mitigate inflammatory responses [42]. TNF-α induces autophagy in various cancer cell types, such as Ewing sarcoma [65], human breast cancer [66], and human T lymphoblastic leukemia cells [67]. The most widely used autophagy marker is LC3, a central protein in the autophagy pathway related to substrate selection and autophagosome biogenesis [37]. Here, the autophagy results were supported by the increased LC3 expression and decreased *TNF* expression in the cells treated with all the concentrations of SORA and DADS alone or in combination. Moreover, there was an upregulation of NRF2 protein levels, which has been associated with the autophagy thought to be involved in the action of p62 [68]. Autophagy is marked by an accumulation of p62, which sequesters Keap1, a negative regulator of Nrf2 [69].

We used the STITCH platform to predict the signaling pathways involved in the effects of DADS and SORA in combination. Genes and proteins are correlated with the MAPK signaling pathway, which affects a wide variety of cellular processes, including proliferation, differentiation, apoptosis, and stress responses [70]. MAPKs are a family of serine/threonine kinases that transmit extracellular signals in response to specific intracellular signaling via the ERK, p38 MAPK, and JNK pathways [71]. Activation of the JNK and p38 MAPK pathways is necessary for programmed cell death, and ERKs are related to cancer cell proliferation and resistance to apoptosis [72]. Some studies have reported that DADS downregulated the MAPK signaling pathway and thereby suppressed the inflammatory responses [73,74]. 

## 5. Conclusions

Taken together, our results demonstrated that SORA and DADS in combination inhibited the cell proliferation, migration, and invasion of hepatocellular carcinoma cells in vitro and modulated the expression of the genes and proteins that suggested the participation of the MAPK pathway. The SORA and DADS combination is a promising therapeutic alternative to develop new clinical protocols to treat patients with hepatocellular carcinoma, especially those with disease in advanced stages.

## Figures and Tables

**Figure 1 pharmaceutics-14-02582-f001:**
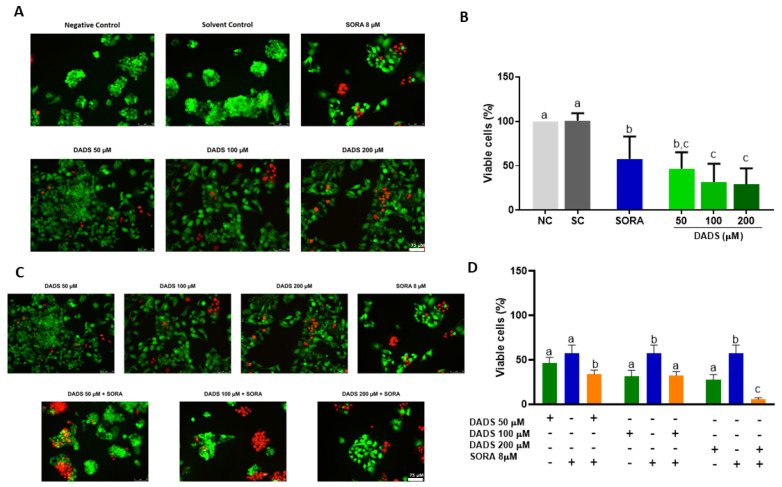
Cell viability of HepG2 cells treated with diallyl disulfide (DADS) and sorafenib (SORA). Representative images of HepG2 cells treated with DADS and SORA, either alone (**A**) or in combination (**C**). Calcein-positive cells (green) are viable; ethidium homodimer-positive cells (red) are dead. HepG2 cells were treated with DADS (50–200 μM) and SORA (8 µM), either alone (**A**,**B**) or in combination (**C**,**D**), for 72 h, and cell viability was assessed using the LIVE/DEAD^®^ assay kit. NC: negative control (culture medium); SC: vehicle control (0.25% DMSO). Results are expressed as mean ± standard deviation (n = 3). Distinct letters indicate statistical significance (*p* < 0.05; ANOVA and Tukey’s post-test). Scale bar: 75 µm.

**Figure 2 pharmaceutics-14-02582-f002:**
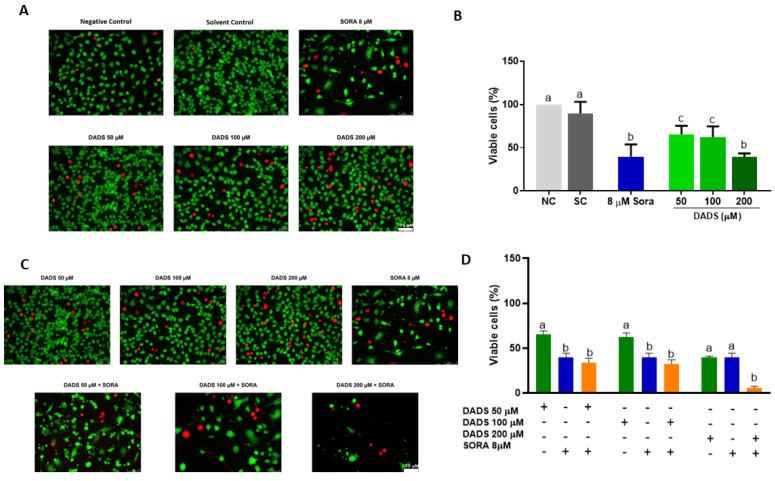
Cell viability of HUVEC cells treated with diallyl disulfide (DADS) and sorafenib (SORA). Representative images of HUVEC cells treated with DADS and SORA, either alone (**A**) or in combination (**C**). Calcein-positive cells (green) are viable; ethidium homodimer-positive cells (red) are dead. HUVEC cells were treated with DADS (50–200 μM) and SORA (8 µM), either alone (**A**,**B**) or in combination (**C**,**D**), for 72 h, and cell viability was assessed using the LIVE/DEAD^®^ assay kit. NC: negative control (culture medium); SC: vehicle control (0.25% DMSO). Results are expressed as mean ± standard deviation (n = 3). Distinct letters indicate statistical significance (*p* < 0.05; ANOVA and Tukey’s post-test). Scale bar: 75 µm.

**Figure 3 pharmaceutics-14-02582-f003:**
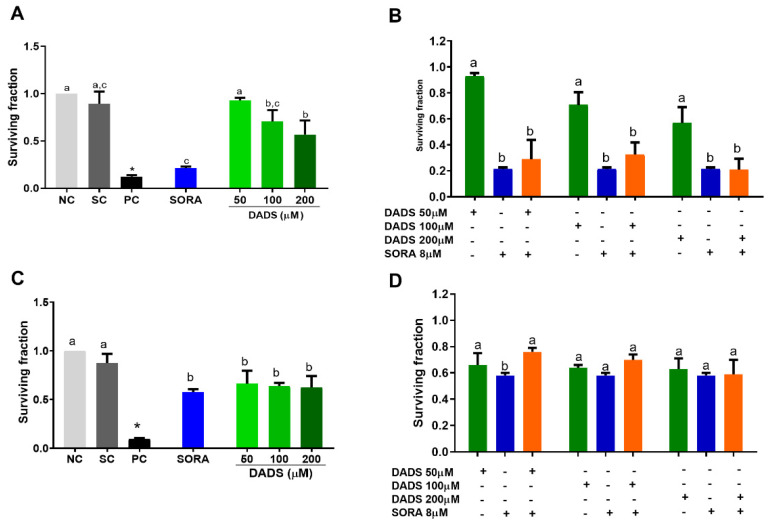
Inhibition of HepG2 and HUVEC cell proliferation by diallyl disulfide (DADS) and sorafenib (SORA). Survival fraction of HepG2 (**A**,**C**) and HUVEC (**B**,**D**) cells treated with DADS and SORA alone (**A**,**B**) or in combination (**C**,**D**) for 72 h and 12 days, as assessed by the clonogenic assay of cell proliferation. NC: negative control (culture medium); SC: vehicle control (0.25% DMSO). Results are expressed as mean ± standard deviation (n = 3). Distinct letters indicate statistical significance (*p* < 0.05; ANOVA and Tukey’s post-test). * *p* < 0.05 vs. negative control (Student’s *t* test).

**Figure 4 pharmaceutics-14-02582-f004:**
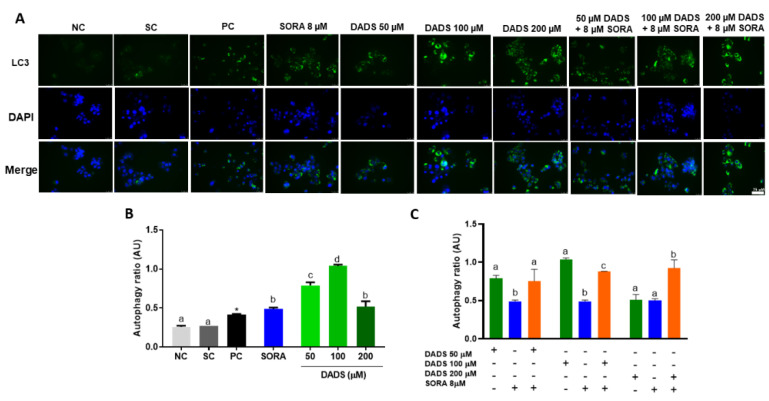
Autophagy-mediated cell death in HepG2 cells induced by diallyl disulfide (DADS) and sorafenib (SORA). Autophagy-induced cell death in HepG2 cells after 24 h of treatment with SORA and DADS, either alone or in combination. (**A**) Photomicrograph of autophagosomes labeled in green and cell nuclei labeled in blue. (**B**) Autophagy induction after treatment with DADS (50, 100, and 200 µM) and SORA (8 µM) alone. (**C**) Autophagy induction after treatment with DADS (50, 100, and 200 µM) and SORA (8 µM) in combination. NC: negative control (culture medium); SC: vehicle control (0.25 % DMSO); PC: positive control (100 μM chloroquine). Mean ± standard deviation (n = 3). Distinct letters indicate significant difference (*p* < 0.05; one-way ANOVA and Tukey’s post-test). * *p* < 0.05 vs. negative control (Student’s *t* test). Scale bar = 75 µm.

**Figure 5 pharmaceutics-14-02582-f005:**
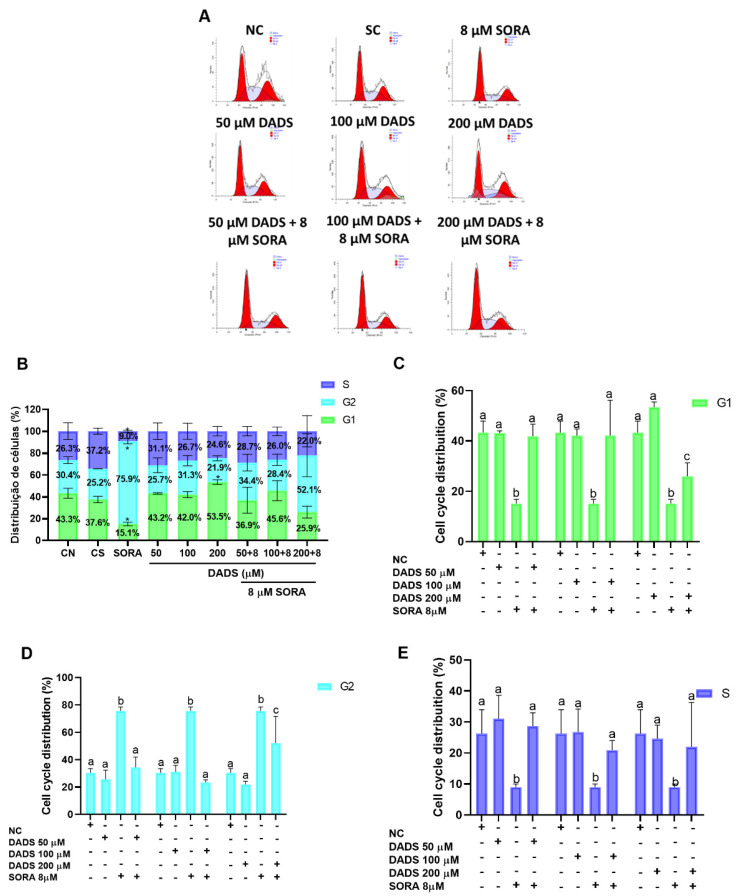
Distribution across the cell cycle phases of HepG2 cells treated with diallyl disulfide (DADS) and sorafenib (SORA). Cell cycle regulation by SORA and DADS after 72 h of treatment, as assessed by flow cytometry. (**A**) Flow cytometry histograms of HepG2 cells treated with DADS and SORA, either alone or in combination, and stained with propidium iodide. (**B**) Cell cycle profile of cells treated with DADS and SORA alone. (**C**–**E**) Distribution of HepG2 cells in the G1 (**C**), G2 (**D**), and S (**E**) phase of the cell cycle after treatment with DADS and SORA, either alone or in combination, and DNA labeling with propidium iodide. NC: negative control (culture medium); SC: vehicle control (0.25% DMSO); SORA: sorafenib; DADS: diallyl disulfide. Distinct letters indicate significant difference within the same group (*p* < 0.05; two-way ANOVA and Tukey’s post-test). Data are expressed as mean ± standard deviation of three independent experiments. * *p* < 0.05 vs. negative control (Student’s *t* test).

**Figure 6 pharmaceutics-14-02582-f006:**
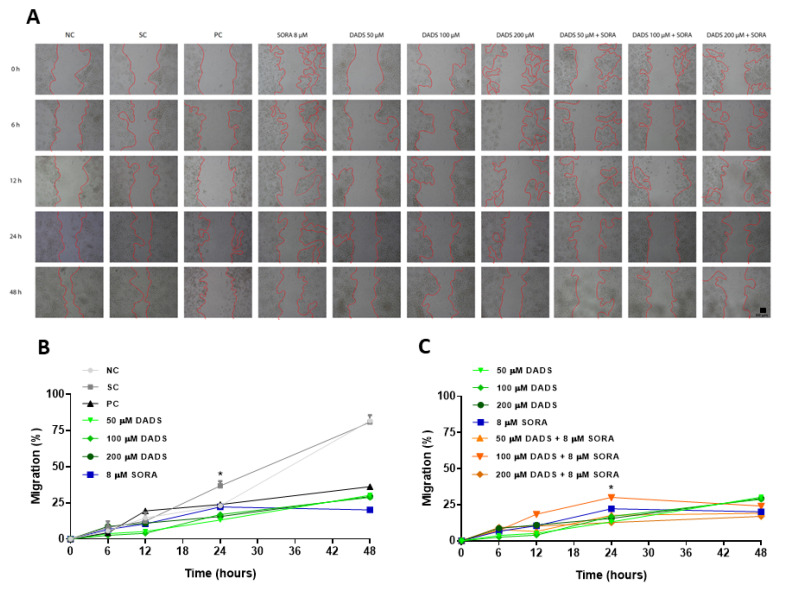
Inhibition of HepG2 cell migration by diallyl disulfide (DADS) and sorafenib (SORA). Representative images from in vitro scratch wound healing assay demonstrating cell migration into the cell-free region (**A**) after treatment with different concentrations of DADS (50, 100, and 200 μM) and SORA (8 μM), either alone or in combination, for 0, 6, 12, 24, and 48 h in HepG2 cells isolated (**B**) and combined treatment (**C**). Wound healing (%) was calculated from the migration distance of HepG2 cells from the control group relative to the treated groups. NC: negative control (PBS); SC: vehicle control (0.25% DMSO); PC: positive control (300 μM MMS). Data are expressed as mean ± standard deviation (n = 3) and analyzed by one-way ANOVA and Dunnett’s post-test (*p* < 0.05). * *p* < 0.05 vs. negative control (Student’s *t* test). Scale bar: 50 µm.

**Figure 7 pharmaceutics-14-02582-f007:**
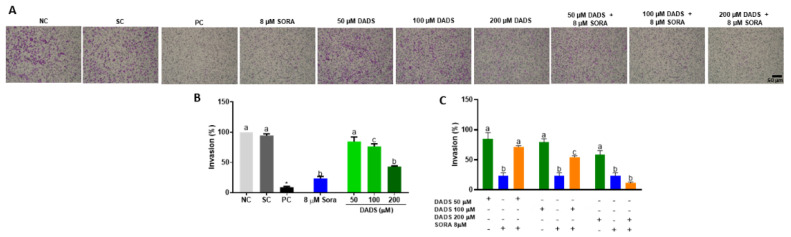
Insert-mediated cell invasion in HepG2 cells treated with diallyl disulfide (DADS) and sorafenib (SORA)*. (***A**) Representative images the adherent cell monolayer of HepG2 cells treated for 48 h with DADS and SORA, either alone or in combination, as recorded with a camera coupled to an inverted microscope. B-C Relative cell migration of HepG2 cells treated with DADS and SORA, either alone (**B**) or in combination (**C**). Cells (%) was calculated from the migration distance of HepG2 cells from the control group relative to the treated groups. NC: negative control (PBS); SC: vehicle control (0.25% DMSO); PC: positive control (300 μM MMS). Data are expressed as mean ± standard deviation (n = 3) and analyzed by one-way ANOVA and Dunnett’s post-test (*p* < 0.05). * *p* < 0.05 vs. negative control (Student’s *t* test). Distinct letters indicate statistical significance. Scale bar: 50 µm.

**Figure 8 pharmaceutics-14-02582-f008:**
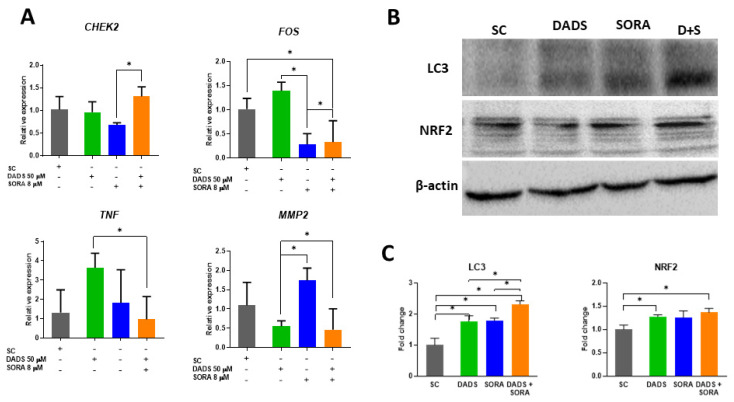
Relative expression of genes and proteins in HepG2 cells treated with diallyl disulfide (DADS) and sorafenib (SORA)*. (***A**) RT-qPCR analysis of the expression levels of *TNF*, *MMP2*, *FOS*, and *CHEK2* genes relative to the reference genes *ACTB*, *GAPDH*, and *HPRT1* in HepG2 cells treated for 24 h with DADS and SORA, either alone or in combination. (**B**) Representative image of Western blot protein bands. (**C**) Fold change of protein bands quantified using the ImageJ software. SC: vehicle control (0.25% DMSO); DADS: 50 µM diallyl disulfide; SORA: 8 µM sorafenib. Representative data from three independent experiments with similar results. * *p* < 0.05 (Student’s *t* test).

**Figure 9 pharmaceutics-14-02582-f009:**
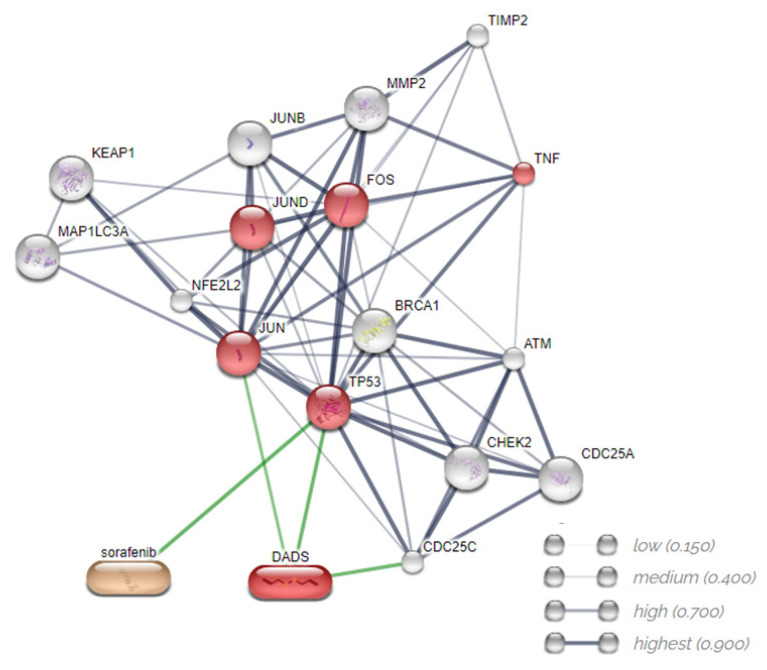
Predicted interactions among diallyl disulfide, sorafenib, genes, and proteins using the STITCH database. Gray lines represent interaction among proteins. Green lines represent interactions between compounds and proteins. Source: STITCH (http://stitch.embl.de/cgi/network.pl?taskId=nv9TPmaTyMjt (accessed on 30 August 2022).

## Data Availability

Not applicable.

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
