# Peer review of "Diallyl Disulfide Induces Chemosensitization to Sorafenib, Autophagy, and Cell Cycle Arrest and Inhibits Invasion in Hepatocellular Carcinoma"

_pharmaceutics, 2022, doi:10.3390/pharmaceutics14122582_

Round 1

Reviewer 1 Report

The Paper is well written and accepted in the present form. 

Author Response

We would like to thank the reviewer 1 who accepted the manuscript without reviews.

Reviewer 2 Report

Machado et al. examined the effect of DAD and SORA either alone or in combination for the treatment of hepatocellular carcinoma. They performed lots experiments to meet their hypothesis. However, there are major concern about the dose selection and subsequent experiments.   

Major revisions:

It is not clear how the authors choose the doses for the combination of DAD and SORA. A better approach would be calculating the combination index as described in the paper below. Moreover, the authors should describe the rationale for the selection of 50, 100 and 200 µM DAD and 8 µM SORA.

 https://aacrjournals.org/clincancerres/article/10/23/7994/185407/Evaluation-of-Combination-ChemotherapyIntegration

If the combination of DAD and SORA is cytotoxic to both HepG2 and HUVEC cells, how can it be clinically relevant for the treatment of hepatocellular or other cancers?

With the lowest dose of DAD (50 µM) plus 8 µM SORA, less than 50% cells are viable for both HepG2 and HUVEC cells, why the authors selected this combination for subsequent experiments?

The doses used for invasion and migration assays are too high. Normally, the IC25 of drugs should be used for such assays. Otherwise, the cells may not be able to invade or migrate since more than 50% will die at IC50 dose and no cells or only few cells will survive at 100 or 200 µM DAD plus 8 µM SORA.

In figure 5, the original images of cell cycle distribution should be included beside the bar diagram.

Minor revisions:

Please revise this sentence “A promising strategy to treat cancer is the association between chemotherapeutics and bioactive compounds isolated from plants”. There are lots of typographical and grammatical errors in the manuscript. The authors should revise.

Reviewer 3 Report

This paper investigates the effects of combined treatments with Diallyl disulfide (DADS) and sorafenib (SORA) on hepatocellular carcinoma cells. The results highlighted some enhanced effects of combination treatments, compared to single compounds treatments, and proposed indicative mechanisms of action. However, this manuscript has several lacks, mostly due to imprecisions and superficial conclusions. It needs to be deeply revised and implemented both in Results and Discussion sections. The weaker portions are those regarding the results about modulation of cell cycle and modulation of specific proteins expression, together with the speculation on the mechanisms of action. To follow you’ll find a list of comments point by point and in attachment you’ll find a revised copy of your manuscript reporting several of my notes.

11.      In general, the Results section should be more accurate, introducing the data with aims and motivations of the performed analysis. Moreover, some data are shown in the figures and not described in the text, some data in the graphics do not match, as well as some results reported in the text do not match data in the graphs, some figure panels do not correspond to those mentioned in the text, few graphics in the figures are not formatted, few Spanish words, no scale bars in the microscope images … you’ll find the details in the revised manuscript;

22.      Paragraph 3.4 DADS and SORA combination modulates cell cycle: To my opinion it should be necessary to show the control results, that is: how is the cell cycle distribution for HepG2 untreated cells? This is mandatory to make a comparison and to conclude that the combined treatment “modulates cell cycle”, comparison with single compounds treatments is not indicative. Moreover, I found the representation of the cell cycle results in Figure 5 a little confusing. Personally, I like better to view all the three cell cycle phases in the same graph, having one graph for each treatment, I believe that the results could so appear clearer.

33.       Paragraph 3.6 DADS and SORA combination induces expression of genes and proteins related to cell death pathways: Here, in the title of this paragraph, the authors imply that the expression of such genes/proteins was upregulated by the combined treatment, instead they showed and described that the expression of all TNF, MMP2 and FOS proteins (or genes?)  was downregulated. This point must be clarified. To my opinion, all this paragraph is weak because the authors did not introduce the kind of analysis they performed and the reason why they analyzed those particular genes/proteins. In the Material and Methods section they list a bunch of genes which expression should have been tested by RT-PCR but they never showed the results. Moreover, not all the results shown are described/commented in the text and it is not clear if or which of them were obtained by RT-PCR or Western Blotting.

The STITCH analysis should be better described and commented (as well the Figure 9), explaining the parameters used and the meaning of obtained results, which are only predictive.

44.       Discussion must be improved in accordance to a better description of the Results. Some proteins are cited without clarifying their role in cell proliferation, invasion and migration (MMP2, FOS, CHEK2, TNF-α). The increased expression of CHEK2, that has not been mentioned in the Results, has been correlated with the block of cell cycle in G1 phase; this relation should be elaborated on, in order to come to the conclusion that “…the high FOS expression followed by induction of cell cycle arrest blocked the process and inhibited cell proliferation…” (lines 402-402). To me this might be just a supposition to be discuss. Also, the expression of FOS was described as downregulated just few lines above (396-397). Discussing the STITCH analysis results the authors argument about a predicted involvement of the MAPK pathway but they haven’t mention it in the Results section. This topic can be very interesting if deepened and I suggest to investigate the levels of expression/activation for at least one of MAPK pathway kinases (for example ERK and phospho-ERK) to verify this hypothesis.

As I mentioned above, more specific notes are given in the revised version of the manuscript.

I hope that my comments could be helpful for the authors in improving their valuable work with the aim that after a deep revision it could be accepted for publication in the Pharmaceutics Journal.

Round 2

Reviewer 3 Report

The manuscript has been consistently improved in this revised version. I made just few comments which the authors can find as notes in the revised copy of the manuscript-v2. Thank you to the authors for their appropriate reply.

I recommend the publication of this paper after minor revisions.
